# Disease-modifying therapies and cost-of-illness progression among people newly diagnosed with multiple sclerosis: a national register-based cohort study covering treatment initiation with interferons, glatiramer acetate or natalizumab

Korinna Karampampa [ID],[1] Hanna Gyllensten [ID],[2] Emilie Friberg,[1] Chantelle Murley [ID],[1] Andrius Kavaliunas,[1] Jan Hillert [ID],[1] Tomas Olsson,[1] Kristina Alexanderson[1]

[1]Department of Clinical Neuroscience, Karolinska Institute, Stockholm, Sweden
[2]Institute of Health and Care Sciences, University of Gothenburg, Goteborg, Sweden

**Correspondence to**
Dr Korinna Karampampa;
korinna.karampampa@ki.se

## ABSTRACT

**Objectives** Disease-modifying therapies (DMTs) can slow disease progression in multiple sclerosis (MS). The objective of this study was to explore the cost-of-illness (COI) progression among newly diagnosed people with MS in relation to the first DMT received.

**Design and setting** A cohort study using data from nationwide registers in Sweden.

**Participants** People with MS (PwMS) in Sweden first diagnosed in 2006–2015, when aged 20–55, receiving first-line therapy with interferons (IFN), glatiramer acetate (GA) or natalizumab (NAT). They were followed up through 2016.

**Outcome measures** Outcomes (in Euros, €) were: (1) secondary healthcare costs: specialised outpatient and inpatient care including out-of-pocket expenditure, DMTs including hospital-administered MS therapies, and prescribed drugs, and (2) productivity losses: sickness absence and disability pension. Descriptive statistics and Poisson regression were computed, adjusting for disability progression using the Expanded Disability Status Scale.

**Results** 3673 newly diagnosed PwMS who were treated with IFN (N=2696), GA (N=441) or NAT (N=536) were identified. Healthcare costs were similar for the INF and GA groups, while the NAT group had higher costs (p value<0.05), owing to DMT and outpatient costs. IFN had lower productivity losses than NAT and GA (p value>0.05), driven by fewer sickness absence days. NAT had a trend towards lower disability pension costs compared with GA (p value>0.05).

**Conclusions** Similar trends over time for healthcare costs and productivity losses were identified across the DMT subgroups. PwMS on NAT maintained their work capacity for a longer time compared with those on GA, potentially leading to lower disability pension costs over time. COI serves as an objective measure to explore the importance of DMTs in maintaining low levels of progression of MS over time.

## STRENGTHS AND LIMITATIONS OF THIS STUDY

⇒ Data from nationwide registers with nationwide coverage were used, preventing recall and selection bias in the productivity loss and healthcare cost calculations.

⇒ Additional costs (primary healthcare, rehabilitation, home help, home investments to improve mobility, other out-of-pocket expenses, sickness absence spells of 14 days or less, productivity reductions while being present at work and informal care costs) were unable to be included in our cost-of-illness (COI) calculations due to lack of register information.

⇒ Some factors that could potentially be strongly associated with the COI progression over time after the first-line therapy was stopped were not measured in this study (eg, type of second-line therapy, adherence to treatment).

⇒ While Expanded Disability Status Scale (EDSS) scores were used to adjust the COI progression for imbalances in disability progression across the three treatment groups as per established methodology, EDSS does not capture all relevant multiple sclerosis symptoms that can limit everyday activities, such as fatigue.

⇒ The observational study design prevents any strong conclusions with respect to causality, and limitations of the data set and setting could limit the generalisability of our findings.

## INTRODUCTION

Multiple sclerosis (MS) is a highly disabling chronic neurological disease,[1] usually diagnosed when aged 20–40 years.[1] Sweden has one of the highest prevalence of MS in Europe, estimated at 189 cases over 100 000 individuals.[2]

In 2010, the annual cost-of-illness (COI) of MS in Sweden, from a societal perspective, was approximately €414 million.[3] Productivity losses[3] and the cost of disease-modifying therapies (DMT) were the two main cost drivers identified.[4]

The COI of MS has been shown to increase with disease's disability progression.[4] However, treatment with DMTs has been associated with slowing this progression; plausibly also reducing the increase in the COI in MS over time.[5]

Until now, only a few observational studies have been conducted evaluating the COI progression in MS in relation to DMTs. However, these studies do not have a longitudinal design,[6] and have also not accounted for disability progression.

The aim of this study was to explore the COI progression in MS among newly diagnosed people with MS (PwMS) in Sweden, in relation to the first DMT received (interferons (IFN) vs glatiramer acetate (GA) vs natalizumab (NAT)), adjusting for disability progression over time.

## METHODS
### Study design and data sources
This was a register-based prospective cohort study of PwMS in Sweden who fulfilled the following two inclusion criteria:

► According to the Swedish MS register (SMSreg)[7] had their first MS diagnosis in one of the years 2006–2015 (index year), when aged 20–55 and lived in Sweden when first diagnosed, (maximum age of 55 was chosen to allow follow-up until ordinary retirement age of 65)
► Received IFN, GA or NAT (DMTs licensed for MS treatment in Sweden during the study period), within 1 month after diagnosis, and they stayed on the therapy for at least 6 months.

Never-users of DMTs, those not on IFN/GA/NAT within 1 month from MS diagnosis, and/or not receiving it for at least 6 months after treatment initiation, and those receiving a DMT before MS diagnosis, were not included in the study.

All included PwMS were followed through 2016 in nationwide registers.[7 8] The following anonymised micro-data, kept by four authorities, were used:
► Stockholm Region
  – SMSreg[7]: diagnosis age and date, type of MS, Expanded Disability Status Scale (EDSS) and type(s) and dates of DMT treatment(s).
► National Board of Health and Welfare
  – National Patient Register (NPR)[8]: dates and diagnoses for inpatient and specialised outpatient healthcare (ie, secondary healthcare).
  – Swedish Prescribed Drug Register (SPDR)[8]: dates, names and costs for all prescribed drugs dispensed from pharmacies.
  – Cause of Death Register[8]: year of death.
► Statistics Sweden: Longitudinal Integration Database for Health Insurance and Labour Market Studies[8];

sex, birth year, educational level, country of birth, type of living area, family situation, annual income from work and/or benefits (yes/no).
► Swedish Social Insurance Agency: Micro Data for the Analysis of Social Insurance register[8]; dates, diagnoses and grade (full-time or part-time) of sickness absence (SA) in SA spells >14 days and disability pension (DP).

The registers were linked using the unique personal identity number that all residents in Sweden are assigned.

PwMS who switched from one DMT group to another relevant group before the 6 months cut-off, were allocated to the later DMT group. PwMS were censored at migration or death.

The following sociodemographic characteristics, measured at the year of diagnosis (index year), were considered: sex, age at index year, educational level, country of birth, type of living area and family situation.

Four types of MS at diagnosis were identified, based on information available from SMSreg: Relapsing-Remitting MS, Progressive-Relapsing MS, Primary-Progressive MS and Secondary-Progressive MS (SPMS).

Multimorbidity in the index year was derived using the Rx-Risk Comorbidity Index,[9] based on the type of drugs prescribed to PwMS according to SPDR.

Disability in MS was based on information from the EDSS available. The score of EDSS ranges from 0 to 10, with 0.5 step intervals (0 indicating no impairment, while 10 indicating death from MS).[10] A clinically meaningful change in the EDSS score is a change of at least one point in patients with EDSS <5.5, and 0.5 point for those with EDSS of ≥5.5.[11]

Where multiple EDSS scores for each calendar year were available, the highest EDSS value was retained. In case of missing EDSS scores in SMSreg, the mean annual score was used. This was computed from the scores of PwMS in the same DMT group, index year and with the same type of MS. This method of imputation was chosen to account for individuals who had multiple missing EDSS values during the study follow-up.

### Study outcomes
Healthcare resources consumed by PwMS as well as their SA and DP days were calculated and multiplied with unit costs (table 1) to determine the societal COI of MS. The estimated COI included all such costs occurring for PwMS during a calendar year, irrespective of the main diagnosis behind the cost.

Inpatient and outpatient costs were calculated for each year of follow-up, by multiplying the number of days/visits, derived from NPR, with the cost per 1.0 retrospective nationwide weight from the diagnosis-related groups system. Co-payment costs for these visits were also estimated by summing daily/per visit fees for inpatient and outpatient care. We assumed that the reimbursement periods started on 1 January, so costs exceeding the co-payment ceiling in Sweden for 1 year's period, were set to the maximum amount allowed per year (table 1).

Table 1  Unit costs used in the calculation of healthcare costs and productivity losses

|  | Year | Value in 2022 SEK | Value in 2022 Euros* | Source |
|---|---|---|---|---|
| Average inpatient and outpatient cost per 1.0 DRG | 2006 | 52 721 kr | €4960 | Swedish Association of Local Authorities and Regions (Sveriges Kommuner och Regioner), KPP Somatik vård.[28] |
|  | 2007 | 51 946 kr | €4887 |  |
|  | 2008 | 53 151 kr | €5000 |  |
|  | 2009 | 53 561 kr | €5039 |  |
|  | 2010 | 52 187 kr | €4910 |  |
|  | 2011 | 52 010 kr | €4893 |  |
|  | 2012 | 51 529 kr | €4848 |  |
|  | 2013 | 53 233 kr | €5008 |  |
|  | 2014 | 55 218 kr | €5195 |  |
|  | 2015 | 57 720 kr | €5430 |  |
|  | 2016 | 59 300 kr | €5579 |  |
| Co-payment for hospital stay (cost per day of stay) | 2018 | 104 kr | €10 | Assume 100 SEK per day, as this is the case for the majority of the regions (including Stockholm).[29] The max co-payment amount for inpatient care was set to 1500 SEK per year (assumption for whole Sweden, based on information from the region Västra Götaland).[30] |
| Co-payment for visit in specialised care (cost per visit) | 2018 | 284 kr | €28 | Swedish Association of Local Authorities and Regions (18). The max co-payment amount for outpatient care was set to 1100 SEK per year. Only one region in Sweden has a max co-payment less than 1100 SEK; so it was assumed 1100 SEK for the entire country. Swedish Association of Local Authorities and Regions.[29] |
| Cost per month for natalizumab | 2018 | 1144 kr | €111 | Treatment with natalizumab is every fourth week (ie, one per month)[31]; therefore, the cost per month of natalizumab was assumed to be the price for one dose of natalizumab (pharmacy's retail price),[31] excluding the cost of administration, which is captured in this study as an outpatient visit to the hospital. |
| Cost per month for rituximab | 2018 | 2022 kr | €197 | Rituximab is used off-label in the treatment of MS; therefore, the exact treatment dosing was not available in the Swedish guidelines for MS treatment. A recent published study in Sweden regarding the use of rituximab for PwMS indicated that the drug dose is 500 mg to 1000 mg per treatment regime (here we assumed the mean, that is, 750 mg per treatment regime), and the mean treatment interval for RRMS PwMS was 7.2 months per year.[32] The cost per month was then assumed to be 1.5 times the pharmacy's retail price per 500 mg of rituximab injection, which was taken from fass.se (9703.61 SEK),[33] divided with the frequency of treatment regimes in months (frequency: every 7.2 months).[32] |
| Monthly salary including employer contributions | 2006 | 42 334 kr | €4127 | The average age-adjusted monthly wage (2018 values) for all employment types was retrieved.[34] It was multiplied with the annual employer contributions, available from the Swedish Tax Authority.[35] The final salary was then inflated to 2022 prices using annual Harmonised Indices of Consumer Prices (HICP) for healthcare available from Eurostat.[14] |
|  | 2007 | 42 457 kr | €4139 |  |
|  | 2008 | 41 416 kr | €4037 |  |
|  | 2009 | 46 112 kr | €4495 |  |
|  | 2010 | 45 591 kr | €4444 |  |
|  | 2011 | 46 264 kr | €4510 |  |
|  | 2012 | 46 283 kr | €4512 |  |
|  | 2013 | 46 596 kr | €4542 |  |
|  | 2014 | 47 701 kr | €4650 |  |
|  | 2015 | 52 025 kr | €5072 |  |
|  | 2016 | 49 797 kr | €4854 |  |

*The annual exchange rate for 2022 from SEK to Euros that was used was 10.6296.
DRG, diagnosis-related groups; MS, multiple sclerosis; PwMS, people with MS; RRMS, Relapsing-Remitting MS; SEK, Swedish krona.

The cost of prescribed drugs was derived from the SPDR, based on the pharmacy's retail price of each dispensed drug package, including the co-payment paid by patients. The annual cost of drugs was calculated by summing the costs for all dispensed drugs during each year of follow-up.

The cost of DMTs not available through SPDR, that is, MS drugs administered in the hospital setting and not dispensed by pharmacies (natalizumab and rituximab for

the study period), was calculated using information in the SMSreg. The number of months on these treatments was multiplied with the cost per month of treatment (table 1).

By adding costs of inpatient and outpatient care and co-payments for inpatient and outpatient care and drugs, the total annual healthcare costs per patient were calculated. Productivity losses were estimated based on the net days with SA and/or DP per year of follow-up, using the human capital approach.[12] Net months were multiplied with the age-adjusted mean salary, adding social security contributions made by employers (table 1).

All people living in Sweden with income from work or unemployment benefits are from 16 years of age covered by the public SA insurance system and can be granted SA benefits if their work capacity is reduced due to disease or injury. The employer reimburses income loss during days 2–14 of an SA spell, after which SA benefits are administered by the Swedish Social Insurance Agency; for unemployed individuals, this happens from day 2. Therefore, to prevent bias in relation to employment status, only SA >14 days was included.

All people aged 19–65 years can be granted DP if their work capacity is long-term or permanently reduced.[13] Both SA and DP can be granted for full-time (100%) or part-time (75%, 50% or 25%) of ordinary work hours,[13] making it possible to have both partial SA and DP concurrently. Hence, we combined them to calculate the net days of SA and DP/year.

All costs were inflated to 2022 prices, applying the harmonised index for consumer prices.[14] Costs calculated in Swedish krona (SEK) were then converted to Euros (EUR) in 2022 values (table 1).

### Analyses

Sociodemographic characteristics, multimorbidity, type of MS at diagnosis, MS disability progression, average COI per patient and year, including 95% CIs, were calculated for each of the three treatment groups.

Pearson's $X^2$ test[15] and the likelihood ratio test[15] were used to explore whether the observed sociodemographic and multimorbidity differences across the three DMT groups were statistically significant (p value<0.05).

Two-tailed Student's T-tests, with unequal variance[16] were used to assess the significance (p value<0.05) of the disability and COI changes over time, within each treatment group, compared with baseline values (at diagnosis) and differences across the treatment groups.

A generalised linear model,[17] with Poisson distribution for costs, was used to estimate mean cost per patient per year, and per group, and 95% CI of the estimation. The model was adjusted with the disability progression of PwMS over time, by setting the EDSS variable to the mean EDSS score across all the three groups for that year.

### Patient and public involvement

None.

## RESULTS

In total, 3673 PwMS, receiving their first treatment with IFN (N=2696), GA (N=441) or NAT (N=536) between 2006 and 2015, were included. Sex, age distribution in the groups, educational level, family situation, country of birth, type of living area and multimorbidity (any) differed across the groups (p value<0.05; table 2). NAT had fewer people with any comorbidities (table 2) versus the IFN and GA groups. The baseline EDSS score was similar for the three groups (online supplemental figure 1).

In online supplemental table 1, the unadjusted mean per patient per year COI for MS is presented, over the 10-year follow-up, and for the three treatment groups. Figure 1A,B show the disability-adjusted healthcare costs and productivity losses. Disability-adjusted costs were also computed for all the cost components (online supplemental figure 2A–F).

People on NAT had higher healthcare costs versus the IFN and GA groups (p value<0.05 for both comparisons), driven by the higher drug costs for this group (online supplemental figure 2D). IFN and GA groups had similar healthcare costs and productivity losses (p values>0.05). Productivity losses for NAT versus GA were similar (p value>0.05), while higher when compared with those on IFN (p value>0.05), driven by the higher SA costs for the NAT group (p value=0.055 vs the IFN group). NAT had the lowest DP costs versus the other two groups at the start of follow-up, and there was a trend of lower DP costs compared with those for the GA group (p value>0.05) over time.

## DISCUSSION

In this register-based prospective cohort study we explored the development of the COI of MS over time among newly diagnosed PwMS in Sweden. Costs were estimated in relation to what type of first-line treatment was used; IFN, versus GA, versus NAT, with adjustment for MS disability progression. The analysis subgroups were constructed based on the DMTs that were licensed to be used as first-line therapies for MS in Sweden during 2006 and 2016, and for which we had information during the study's data period (2006–2016).

Overall, the general trends of decreasing healthcare costs and increasing productivity losses that were observed in this study are in line with previous COI studies in MS with a similar longitudinal design.[18 19] No significant differences in healthcare costs were observed between the IFN and GA, while NAT had higher healthcare costs (p value<0.05). This could be explained by the higher treatment cost with NAT,[20] that is, higher drug cost and healthcare usage to administer and monitor PwMS on this DMT. As previous studies have shown, in the last two decades, the cost of drugs has become the main COI driver in MS.[4]

In addition, outpatient costs for PwMS on NAT could have played a role; they were higher for NAT (p

**Table 2** Sociodemographic and multimorbidity characteristics at year of diagnosis (index year)

| | Treatment with interferons n=2696 N (%)* | Treatment with glatiramer acetate n=441 N (%)* | Treatment with natalizumab n=536 N (%)* | Pearson's X² (p value) | Log-likelihood test X² (p value) |
|---|---|---|---|---|---|
| Sex | | | | 7.49 (0.024) | 7.70 (0.021) |
| Women | 1882 (51.24) | 333 (9.07) | 364 (9.91) | | |
| Men | 814 (22.16) | 108 (2.94) | 172 (4.68) | | |
| Age at diagnosis (index year) | | | | 107.26(<0.0001) | 108.56(<0.0001) |
| 20–25 | 321 (8.74) | 45 (1.23) | 114 (3.1) | | |
| 26–30 | 435 (11.84) | 67 (1.82) | 129 (3.51) | | |
| 31–35 | 465 (12.66) | 65 (1.77) | 114 (3.1) | | |
| 36–40 | 477 (12.99) | 88 (2.4) | 74 (2.01) | | |
| 41–45 | 463 (12.61) | 75 (2.04) | 46 (1.25) | | |
| 46–50 | 316 (8.6) | 60 (1.63) | 39 (1.06) | | |
| 51–55 | 219 (5.96) | 41 (1.12) | 20 (0.54) | | |
| Type of MS at diagnosis (index year)† | | | | 2.24 (0.896) | 2.25 (0.896) |
| RRMS | 1960 (65.14) | 302 (10.04) | 419 (13.92) | | |
| PPMS | 23 (0.76) | 4 (0.13) | 4 (0.13) | | |
| SPMS | 187 (6.21) | 31 (1.03) | 33 (1.1) | | |
| PRMS | 31 (1.03) | 6 (0.2) | 9 (0.3) | | |
| Educational level‡ | | | | 72.90 (<0.0001) | 58.09 (<0.0001) |
| Elementary school | 387 (10.84) | 67 (1.88) | 134 (3.75) | | |
| High school | 1290 (36.13) | 210 (5.88) | 225 (6.3) | | |
| University/college | 946 (26.5) | 153 (4.29) | 158 (4.43) | | |
| Country of birth‡ | | | | 17.70 (0.0236) | 14.70 (0.065) |
| Sweden | 2348 (65.77) | 390 (10.92) | 454 (12.72) | | |
| Nordic countries (except Sweden) | 43 (1.2) | 9 (0.25) | 15 (0.42) | | |
| EU27 (except Denmark, Finland and Sweden) | 42 (1.18) | 8 (0.22) | 16 (0.45) | | |
| Rest of the world | 190 (5.32) | 23 (0.64) | 32 (0.9) | | |
| Type of living area | | | | 15.47 (0.004) | 15.30 (0.004) |
| Big cities‡ | 1020 (28.57) | 142 (3.98) | 210 (5.88) | | |
| Medium sized cities | 888 (24.87) | 141 (3.95) | 187 (5.24) | | |
| Rural areas | 715 (20.03) | 147 (4.12) | 120 (3.36) | | |
| Family situation‡ | | | | 140.56 (<0.0001) | 123.74 (<0.0001) |
| Married or cohabitant without children§¶ | 189 (5.29) | 38 (1.06) | 26 (0.73) | | |
| Married or cohabitant with children§¶ | 1021 (28.6) | 165 (4.62) | 117 (3.28) | | |
| Single without children§¶ | 1211 (33.92) | 190 (5.32) | 320 (8.96) | | |
| Single with children§¶ | 174 (4.87) | 28 (0.78) | 25 (0.7) | | |
| Any multimorbidity | | | | 68.35 (<0.0001) | 53.92 (<0.0001) |
| Yes | 2580 (70.24) | 424 (11.54) | 466 (12.69) | | |
| No | 116 (3.16) | 17 (0.46) | 70 (1.91) | | |
| Anxiety/depression | | | | 4.10 (0.129) | 3.91 (0.142) |
| Yes | 137 (3.73) | 28 (0.76) | | | |
| No | 2559 (69.67) | 413 (11.24) | | | |

*The percentages are calculated as n divided with the total number of individuals in the study population (3673 if not otherwise indicated; see †).
†For 18% of the cohort, type of MS information was missing; so, information was available for 3009 PwMS in the study cohort.
‡The total number of individuals with this type of information in the study population (ie, all individuals excluding those with missing information in this category) was 3570.
§Only cohabitants with children in common are registered as cohabitants. Otherwise they were registered as single.
¶Children were considered as being <18 years of age and living at home.
MS, multiple sclerosis; PPMS, Primary-Progressive MS; PRMS, Progressive-Relapsing MS; PwMS, people with MS; RRMS, Relapsing-Remitting MS; SPMS, Secondary-Progressive MS.

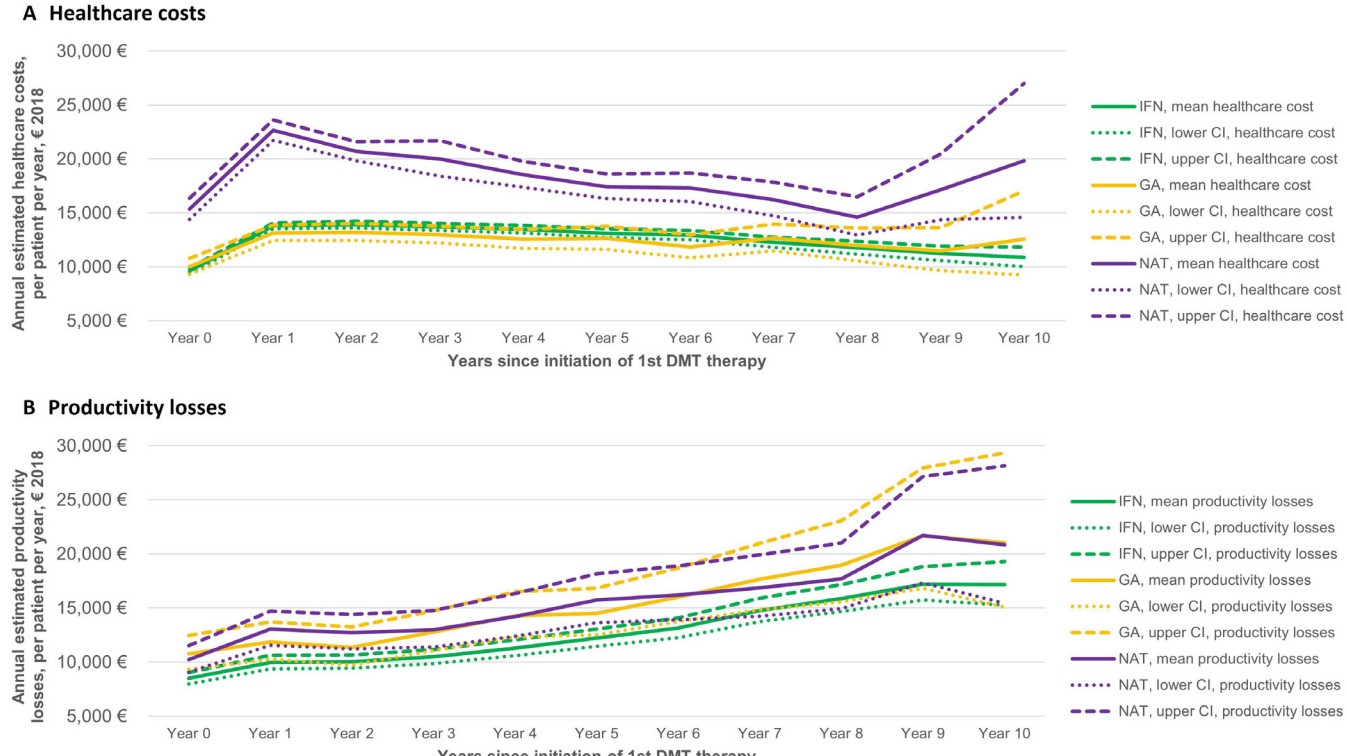

**A  Healthcare costs**

**B  Productivity losses**

**Figure 1** Healthcare costs (A) and productivity losses (B) (estimated mean from the regression) from baseline (year at MS diagnosis) to the end of follow-up, by treatment groups, adjusted for disability progression during the follow-up. DMT, disease-modifying therapy; GA, glatiramer acetate; IFN, interferon; NAT. natalizumab.

value>0.05). For the beginning of follow-up, until PwMS on NAT changed therapy, this could be explained by the fact that these patients require at least one extra outpatient visit per month to get their treatment.[21] Before treatment, additional diagnostic tests (blood tests and in cases MRI scans) have to be performed, not needed for PwMS on other therapies, including IFN and GA.[21] Therefore, when compared with IFN and GA, healthcare costs for NAT could be amplified due to the extra outpatient visits needed for the treatment itself. Had no such visits and tests been necessary, outpatient costs NAT could have been at the same level as the ones for IFN and GA, or even lower.

Furthermore, DMTs were not randomly assigned to PwMS in our observational study, and therefore this study could not consider treatment decisions based on observed disease activity at the time of diagnosis. If treating neurologists assigned NAT to PwMS who could have had potentially higher disease activity, trying to prevent worse disease outcomes, this may have resulted in greater healthcare costs in the NAT group versus IFN or GA, than those that could be expected if DMTs were randomly assigned to PwMS.

Towards the end of follow-up, when people may have switched to other DMTs, or stopped treatment, other factors could be associated with the observed healthcare cost differences, which have not been covered in this study, such as the progression of MS symptoms, relapses and multimorbidity.

Productivity losses for NAT and GA were similar; both had higher productivity losses versus IFN (p value<0.05). Since the comparative effectiveness of IFN and GA is similar,[22] other unmeasured factors than disease progression, such as fatigue and multimorbidity could explain this finding. On the other hand, even if treatment with NAT can lead to lower relapse rates,[23] and hence possibly to slower disease progression, compared with IFN and GA, productivity losses for this group were higher. The need to take time off work when on NAT related to the administration of the therapy (outpatient visits to administer the therapy and run diagnostic tests), and recovery from the administration of this DMT, could explain part of this observed outcome,[21] since it adds extra SA costs for NAT.[24]

In fact, only SA costs were higher for NAT compared with the other groups; DP costs were lower than GA, indicating that even if PwMS receiving NAT as first-line treatment, are having higher MS disability at baseline, and probably higher disease activity compared with those receiving IFN or GA,[21] still, their ability to maintain their long-term work capacity is higher. Experiencing less relapses and in general less MS symptoms like fatigue, which can allow patients to maintain employment longer.[25]

Previously published health economic models presenting the cost progression related to these three DMTs have often been criticised due to their limited data availability as well as the methodological complexities and uncertainties.[25] The COI of MS associated with new

generation therapies, like NAT, compared with older therapies (IFN and GA) has not been studied so far involving longitudinal COI data with multiple years of follow-up for costs and disease progression.

## Strengths and limitations

Main strengths of this study are its prospective cohort design, with several years of follow-up, and that data from high-quality administrative registers could be used. All PwMS in the SMSreg diagnosed with MS in the years 2006–2015 and receiving first-line therapy with NAT, IFN or GA could be followed for several years in nationwide registers, eliminating drop outs and recall biases when using self-reported information from PwMS.[4] However, as there are no nationwide register data in Sweden regarding primary healthcare, rehabilitation, home help, home investments to improve mobility and other out-of-pocket expenses this could not be included in our COI calculations. In addition, this study does not take into account the societal costs associated with premature death due to the disease. Hopefully future studies can include such information, to analyse how MS symptoms and multimorbidity might be associated with the healthcare costs and overall societal burden in MS, based on the DMT treatment administered.

In addition, the longest follow-up time we had was 10 years after diagnosis; while this time can give important information regarding the COI progression for the three treatment groups, only very few patients had this many follow-up years. Therefore, the longer the follow-up, the higher the uncertainty is for the estimated COI presented. Moreover, while the potential effect of the first-line therapy on the COI can be observed for the time PwMS are receiving that treatment and maybe for a short period of time afterwards, also other factors could be strongly associated with the COI progression over time after the first-line therapy was stopped, not measured in this study (eg, type of second-line therapy, adherence to treatment).

Another important strength is that robust information on productivity losses was available enabling us to apply the societal perspective on the COI calculations. Productivity losses are the main long-term driver of the COI in MS as shown in this and previous studies.[4] However, short-term SA-spells of 14 days or less were not quantified possibly leading to underestimation of the productivity losses. Short-term SA spells have not been considered for both employed (including self-employed) and unemployed patients, hence no bias based on employment status has been introduced in the calculation of productivity losses.

Moreover, any productivity reductions while being present at work that could potentially be related to the presence of MS were not quantified. Furthermore, productivity losses incurred by partners of PwMS, that is, informal care costs, were not included in the COI calculations; while measuring these costs was beyond the aim

of this study, they are an important cost component when defining the overall COI of MS.[4]

Similar to what previous COI in MS studies have done,[4] we used the EDSS score to explore disability progression for PwMS in the three treatment groups. However, when EDSS information was missing, information had to be imputed. This has likely diluted the EDSS score used in the adjusted COI analysis. In addition, EDSS has been primarily developed and used for research purposes, not for clinical or treatment decisions in MS, and its scores are less effectively tracking disease progression over time.[26] Also, some MS symptoms, like fatigue, which are important and can limit the everyday activities of PwMS early in the disease trajectory, are not well detailed in the EDSS assessment,[26] obtaining in this way a skewed result regarding the disability of PwMS. Additional clinical measures could have complemented our analysis.

Another limitation is that the generalisability of our findings can be limited. The study cohort was taken from SMSreg, which back in 2006 had approximately 50% coverage of all PwMS in Sweden, and 80% in 2015.[27] Assuming no distinct differences in the sociodemographic characteristics, multimorbidity and disability with the PwMS that our study did not include, our findings could be generalisable to all PwMS in Sweden. Still, in our cohort, an elevated number of people with SPMS at diagnosis was included versus the expected distribution of the type of MS at diagnosis, which could potentially hinder the generalisability of our findings. In addition, generalisation of our findings to other countries is not possible, considering the differences in the organisation of healthcare and social security systems.

## Conclusions

Using real-world data from Swedish nationwide registers, this study explored the association between the type of first-line DMT received (IFN vs GA vs NAT) with COI progression. The findings of a trend towards lower DP costs indicated that PwMS on NAT could potentially maintain their work capacity for a longer time. COI serves as an objective measure showing the importance of the use of second-generation treatments in maintaining at low levels the progression of MS over time, and hence the overall COI.

**Contributors** All authors (KK, HG, EF, CM, JH, AK, TO, KA) contributed to the conceptualisation of the research questions, the study design and methods. KK performed the analysis of the data, interpreted the study findings and drafted the manuscript, and is the main author responsinble for the overall content of this research. All remaining authors (HG, EF, CM, JH, AK, TO, KA) assisted in the interpretation of the study findings and contributed with comments/suggestions and text to the manuscript.

**Funding** This study was supported by an unrestricted investigator-initiated trial grant from Biogen to support the conduct of this study (Award number N/A). We used data from the REWHARD consortium supported by the Swedish Research Council (grant number 2021-00154).

**Competing interests** All authors (KK, HG, EF, CM, JH, AK, TO, KA) are employed or affiliated at the Department of Clinical Neuroscience, Karolinska Institutet, Stockholm, Sweden. KK is currently employed by Celgene/Bristol Myers Squibb; she initiated this study while being employed at Karolinska Institutet

(employment ended in October 2019); since then, she has received no salary from Karolinska Institutet or other type of funding for this research. HG is currently employed part-time by Statfinn/EPID Research (which is part of IQVIA); both companies are contract research organisations that perform commissioned pharmacoepidemiological studies, and therefore are collaborating with several pharmaceutical companies. CM since submission of this paper has begun employment with Macanda AB. AK is currently also employed by Takeda Pharma AB. JH, KA and EF are collaborating with several pharmaceutical companies; EF has received an unrestricted MS research grant from Celgene/Bristol Myers Squibb. TO has received advisory board and/or lecture honoraria, and unrestricted MS research grants from Biogen, Novartis, Sanofi, Merck and Roche.

**Patient and public involvement** Patients and/or the public were not involved in the design, or conduct, or reporting, or dissemination plans of this research.

**Patient consent for publication** Not applicable.

**Ethics approval** The project received ethical approval from the Regional Ethical Review Board in Stockholm. Approval numbers: 2007/762-31; 2009/23-32; 2009/1917-32; 2010/466-32; 2011/806-32; 2011/1710-32; and 2014/236-32.

**Provenance and peer review** Not commissioned; externally peer reviewed.

**Data availability statement** Data may be obtained from a third party and are not publicly available. The data used in this study is administered by the Division of Insurance Medicine, Karolinska Institutet, and cannot be made publicly available. According to the General Data Protection Regulation, the Swedish law SFS 2018:218, the Swedish Data Protection Act, the Swedish Ethical Review Act and the Public Access to Information and Secrecy Act, these type of sensitive data can only be made available, after legal review, for researchers who meet the criteria for access to this type of sensitive and confidential data. Readers may contact Professor Kristina Alexanderson (kristina.alexanderson@ki.se) regarding the data.

**ORCID iDs**
Korinna Karampampa http://orcid.org/0000-0002-2578-1865
Hanna Gyllensten http://orcid.org/0000-0001-6890-5162
Chantelle Murley http://orcid.org/0000-0003-4150-4275
Jan Hillert http://orcid.org/0000-0002-7386-6732

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
