## [Reviewer comments · BMJ Open]

ARTICLE DETAILS

TITLE (PROVISIONAL)	Disease-modifying therapies and cost-of-illness progression among people newly diagnosed with multiple sclerosis: a national register-based cohort study covering treatment initiation with interferons, glatiramer acetate, or natalizumab
AUTHORS	Karampampa, Korinna; Gyllensten, Hanna; Friberg, Emilie; Murley, Chantelle; Kavaliunas, A; Hillert, Jan; Olsson, Tomas; Alexanderson, Kristina

VERSION 1 – REVIEW

REVIEWER	Taylor, Bruce University of Tasmania, Menzies Research Institute
REVIEW RETURNED	04-Oct-2022

GENERAL COMMENTS	This is an interesting assessment of the COI associated with MS drugs at diagnosis in a large Swedish cohort diagnosed between 2006 and 2016. This is actually not a full COI study as it only uses available registry data and is not a complete COI study as there is no primary care data etc this is acknowledged by the authors but this fact could be more prominent in the abstract. The authors should also acknowledge potential and likely significant bias in the prescription of drugs. as they were not randomly assigned and it is likely that those thought by treating neurologists to have a potentially worse outcome would be assigned to natalizumab rather than the other drugs this may result in greater health care costs in the NAT group than would be expected if they were randomly assigned. Also how can 250 people have SPMS at diagnosis this doesn't make a lot of sense. Also when imputing EDSS scores would it not be better to do this at the individual level rather than take a mean value from the whole cohort. In the abstract the authors state that DP costs were lower for NAT than GA where the actual numbers are nearly identical and the graphs overlap almost completely.
--

REVIEWER	Amirsadri, Mohammadreza Isfahan University of Medical Sciences, Clinical pharmacy and pharmacy practice
REVIEW RETURNED	15-Jan-2023

GENERAL COMMENTS	1- Please change conclusion of the abstract, as it neither summarize
--

	the key supporting ideas you discussed nor offer your final impression on the central idea. 2- P: 6: “The cohort of PwMS were followed until maximum 2016 in nationwide registers “ Is it a prevalence-based or an incidence-based COI study? Please clarify it. If follow up period has not been the same for the evaluated patients, how you justify it? (e.g. some patients from 1015 to 2016 and some from 2006 to 2016). 3- P:7: “PwMS who switched from one DMT group to another relevant group before the 6-months cut-off, were allocated to the later DMT group.” This could affect the results of the study as the evaluated effects have not been related to one specific drug. 4- P:7: “In analyses, PwMS were censored at migration or death.” So how premature death due to the disease has been valued? 5- P: 9: “The employer usually reimburses income loss during days 2-14, after which SA benefits are administered by the Swedish Social Insurance Agency; for unemployed individuals, this happens from day 2. Therefore, to prevent bias in relation to employment status, only SA >14 days was included.” As the evaluation has been performed from social perspective, all costs, including those paid by the employers, should have been taken into account. 6- P9: “All costs were inflated to 2018 prices” Please update to 2022. 7- P:17: “The analysis sub-groups were constructed based on the DMTs that were licensed to be used as 1st line therapies for MS in Sweden during 2006 and 2016” 2006-2016.
--	--

VERSION 1 – AUTHOR RESPONSE

Reviewer’s comments and responses

Reviewer 1: Dr. Bruce Taylor, University of Tasmania

Comment 1: This is an interesting assessment of the COI associated with MS drugs at diagnosis in a large Swedish cohort diagnosed between 2006 and 2016. This is actually not a full COI study as it only uses available registry data and is not a complete COI study as there is no primary care data etc. This is acknowledged by the authors but this fact could be more prominent in the abstract.

Author’s answer: We thank the reviewer for recognizing the contribution of our study to the literature. Indeed, there is no nationwide register of primary healthcare data in Sweden, why we only could use information on secondary healthcare use. We have now made this point clearer in the abstract, clarifying that it is secondary healthcare costs that was collected.

METHODS:

Comment 2: The authors should also acknowledge potential and likely significant bias in the prescription of drugs. as they were not randomly assigned and it is likely that those thought by treating neurologists to have a potentially worse outcome would be assigned to natalizumab rather than the

other drugs this may result in greater health care costs in the NAT group than would be expected if they were randomly assigned.

Author's answer: Thank you for this valid observation. We have now further clarified this limitation in the revised discussion section of the manuscript.

Comment 3: Also how can 250 people have SPMS at diagnosis this doesn't make a lot of sense.

Author's answer: Thank you for pointing this out. When looking at type of MS at diagnosis, approximately 10% of new MS diagnosis appearing in SMreg in our cohort were registered as people having secondary progressive MS. Although this percentage is low, still it can be considered elevated, given the expected disease characteristics at diagnosis. This fact could be attributed to how diagnosis is set, i.e. there is some wait time from disease onset to diagnosis. Therefore, at the time of diagnosis, the disease could be progressed.

We want to emphasize that individuals were selected in our cohort based on when we could identify their first MS diagnosis in the Swedish MS register, and when they have received their first DMT (those who received IFN, GA, or NAT within 1 month after diagnosis and for at least 6 months). Never-users of DMTs, those not on IFN/GA/NAT within 1 month from MS diagnosis, and/or not receiving it for at least 6 months after treatment initiation, and those receiving a DMT before MS diagnosis were excluded from our study, in an effort to capture incident MS cases, who received their first DMT around the same time.

Nevertheless, the presence of a high percentage of people with SPMS at diagnosis can be seen as a limitation for the generalization of the study outcomes, which we now emphasized at the revised discussion section of our manuscript. It does not impact the comparison of the COI across the DMTs, as no statistically significant difference in the percentage of people with SPMS across the three treatment groups was detected.

Comment 4: Also when imputing EDSS scores would it not be better to do this at the individual level rather than take a mean value from the whole cohort.

Author's answer: Thank you for this comment. This method of imputation was chosen in order to count for individuals who had multiple missing EDSS values. We have now clarified this in the revised manuscript.

Comment 5: In the abstract the authors state that DP costs were lower for NAT than GA where the actual numbers are nearly identical and the graphs overlap almost completely.

Author's answer: While there was some numeric difference in the disability pension costs (lower for NAT vs GA), the difference was not statistically significant. We have now added this clarification in the abstract and manuscript.

Reviewer 2: Dr. Mohammadreza Amirsadri, Isfahan University of Medical Sciences

Comments to the Author

Comment 1: Please change conclusion of the abstract, as it neither summarize the key supporting ideas you discussed nor offer your final impression on the central idea.

Author's answer: This is now revised, thank you for your observation.

Comment 2: The cohort of PwMS were followed until maximum 2016 in nationwide registers “ Is it a prevalence-based or an incidence-based COI study? Please clarify it. If follow up period has not been the same for the evaluated patients, how you justify it? (e.g. some patients from 1015 to 2016 and some from 2006 to 2016).

Author's answer: Thank you for this comment. This was an incidence-based cohort; we included individuals with first-time MS diagnosis, who had received treatment from one of the three DMT groups within a month from diagnosis, and had who stayed on it for at least for 6 months. All these individuals were followed through 2016. As the year for inclusion in the cohort differed, the follow-up time differed.

For this reason, results for each of the three therapy groups, and for the comparison, are always averaged in Year 0, Year 1, etc. after index year (the year of the MS diagnosis). Therefore, the same follow-up time is considered to compare outcomes over time. We clarified this further in the manuscript and we also dedicated a paragraph in the discussion section of the manuscript to present the limitation of the lack of identical follow-up time for the entire cohort.

Comment 3: PwMS who switched from one DMT group to another relevant group before the 6-months cut-off, were allocated to the later DMT group.” This could affect the results of the study as the evaluated effects have not been related to one specific drug

Author's answer: Thank you for your comment. The effect of a therapy, from these three category groups, who got switched quickly after its initiation was hypothesized to be unlikely to be connected with the long-term COI we studied. Also, as we stated in the discussion section of our manuscript, while the impact of the first-line therapy on the COI can be observed for the time PwMS are receiving that treatment and maybe for a short period of time afterwards, also other factors could be strongly associated with the COI progression over time after the first-line therapy was stopped, not measured in this study (e.g., type of second-line therapy, adherence to treatment, etc.). Therefore, we applied this hypothesis into our analysis. Given that the focus of this study is the comparison across the three groups, and the same hypothesis was applied to all the three treatment groups, we do not believe that any COI detected differences across the groups would be attributable to this.

Comment 4: In analyses, PwMS were censored at migration or death.” So how premature death due to the disease has been valued?

Author's answer: We did not study the impact of premature death on the COI outcomes. In addition, we did not have any information in our dataset to evaluate whether deaths were attributable to the disease or not. The focus of this study was the comparison of the COI across the three treatment groups, and differences in censoring due to deaths across the groups were not present; therefore, we do not expect any impact of deaths on the comparison of the COI outcomes.

Comment 5: The employer usually reimburses income loss during days 2-14, after which SA benefits are administered by the Swedish Social Insurance Agency; for unemployed individuals, this happens from day 2. Therefore, to prevent bias in relation to employment status, only SA >14 days was included.” As the evaluation has been performed from social perspective, all costs, including those paid by the employers, should have been considered.

Author’s answer: Thank you for your comment. Of course, we would have preferred to have access to data on all short sickness absence spells. However, there is no Swedish nationwide register keeping such data. We used the MiDAS register for information regarding sickness absence and disability pension benefits paid by the public Swedish Social Insurance Agency. As the Swedish Social Insurance Agency does not pay benefits for the first 14 days of a sickness absence spell (but the employer does) for people in paid work, the register does not contain any information on spells ending before day 15 among employed. However, the Swedish Social Insurance Agency cover also the first two weeks during a sickness absence spell for unemployed individuals. In order not to introduce bias in the analyses, we excluded information on these short spells (i.e., among unemployed), as is commonly done in these types of studies.

Moreover, although many sickness absence spells are short, these short spells do not contribute much to the total number of sickness absence days. According to one study about this (unfortunately only published in Swedish) short spells (<15 days) contributed less than 20% of all SA days during a year.

This lack of information is a limitation, and that we can only study more “long-term” productivity losses. This is diligently explained in the

limitation section of our discussion and we also tried to further clarify it. This study aims to study the COI from a societal perspective, and we do so by including productivity losses in the COI. Any transfer payments (i.e., employer compensation or sick-leave payment from the Social Insurance Agency to individuals) are of course not included in our COI calculation, as per established methodology to calculate the COI. As it is stated in our manuscript, we based the productivity losses calculations on net days of sickness absence and disability pension, multiplied with the calculated daily salary following the widely used human capital approach.

Comment 6: All costs were inflated to 2018 prices” Please update to 2022.

Author’s answer: The analysis is now updated (in the text, tables, and figures).

Comment 7: The analysis sub-groups were constructed based on the DMTs that were licensed to be used as 1st line therapies for MS in Sweden during 2006 and 2016” 2006-2016.

Author’s answer: Indeed, this is the case. We have now clarified this further in the revised manuscript.

VERSION 2 – REVIEW

REVIEWER	Taylor, Bruce University of Tasmania, Menzies Research Institute
----------	---

REVIEW RETURNED	27-Mar-2023
GENERAL COMMENTS	Thankyou for addressing my comments so thoroughly
REVIEWER	Amirsadri, Mohammadreza Isfahan University of Medical Sciences, Clinical pharmacy and pharmacy practice
REVIEW RETURNED	09-Apr-2023
GENERAL COMMENTS	As "From disease onset, pwMS survived 14.6 years shorter than the general population ($p<0.001$)." (Willumsen JS, et al. J Neurol Neurosurg Psychiatry 2022;93:1154–1161. doi:10.1136/jnnp-2022-329169) ignoring the impact of premature death could have a great impact on a study, performed from the societal perspective. This should be clearly pointed out as a weakness of the study.

VERSION 2 – AUTHOR RESPONSE

Reviewer’s comments and responses

Reviewer 2: Dr. Mohammadreza Amirsadri, Isfahan University of Medical Sciences

Comment 1: As "From disease onset, pwMS survived 14.6 years shorter than the general population ($p<0.001$)."
(Willumsen JS, et al. J Neurol Neurosurg Psychiatry 2022;93:1154–1161 PubMed . doi:10.1136/jnnp-2022-329169) ignoring the impact of premature death could have a great impact on a study, performed from the societal perspective. This should be clearly pointed out as a weakness of the study.

Author’s answer: We thank the reviewer for pointing out this limitation, we, we have now included relevant text in the strengths and limitations section of the discussion.

VERSION 3 – REVIEW

REVIEWER	Amirsadri, Mohammadreza Isfahan University of Medical Sciences, Clinical pharmacy and pharmacy practice
REVIEW RETURNED	01-May-2023
GENERAL COMMENTS	Although considering loss of productivity due to premature death from social perspective could be valuable, however as it is now mentioned as a limitation of the study, the manuscript can be accepted for publication.